# Non-Invasive Metabolic and Structural Retinal Markers in Patients with Giant Cell Arteritis and Polymyalgia Rheumatica: A Cross-Sectional Study

**DOI:** 10.3390/metabo12090872

**Published:** 2022-09-16

**Authors:** Simon J. Lowater, Torkell J. Ellingsen, Jens K. Pedersen, Jimmi Wied, Jakob Grauslund, Keld-Erik Byg

**Affiliations:** 1Research Unit of Ophthalmology, Department of Ophthalmology, Odense University Hospital, 5000 Odense C, Denmark; 2Research Unit of Rheumatology, Department of Rheumatology, Odense University Hospital, 5000 Odense C, Denmark; 3Department of Clinical Research, University of Southern Denmark, 5000 Odense C, Denmark; 4Department of Medicine, Rheumatology Section, Odense University Hospital, Svendborg Sygehus, 5700 Svendborg, Denmark; 5Department of Neurology, Odense University Hospital, 5000 Odense C, Denmark

**Keywords:** giant cell arteritis, polymyalgia rheumatica, retinal vessel, retinal vasculature, oximetry, retina

## Abstract

Giant cell arteritis (GCA) is a potential sight-threatening disease. Although it is associated with polymyalgia rheumatica (PMR), visual loss is not common in PMR. A retinal oximeter can be used to conduct a direct, non-invasive, in vivo assessment of the vascular system. In a cross-sectional study, we measured the retinal oxygen saturation and retinal vessel calibers in GCA patients, PMR patients, and control participants. Twenty GCA patients (38 eyes), 19 PMR patients (33 eyes), and 12 controls (20 eyes) were investigated. Images were analyzed using Oxymap Analyzer software 2.5.0 (Oxymap ehf., Reykjavik, Iceland). Groups were compared using an age- and sex-adjusted linear mixed model regression. The median (IQR) age for GCA patients was 69.0 (66.5–76.5) years, for PMR 69.0 (67.0–72.0) years, and for the controls 75.5 (71.5–81.0) years, respectively. As compared to the controls (115.3 µm), the retinal arterioles were significantly wider in patients with GCA (124.4 µm; *p* = 0.023) and PMR (124.8 µm; *p* = 0.049). No difference was found in the retinal venular caliber or vascular oxygen saturation. These results indicate that GCA and PMR patients differ similarly in the retinal arteriolar diameter compared to controls. Further studies are needed in order to clarify the underlying inflammatory mechanisms in retinal arteriolar vessels and if these parameters can be used to predict clinical outcomes.

## 1. Introduction

Giant cell arteritis (GCA) is a type of medium and large blood vessel vasculitis with a high risk of ocular morbidities, such as visual impairment [1]. In contrast, polymyalgia rheumatica (PMR) is not considered a type of vasculitis, but a syndrome associated with pain and stiffness, usually in the neck, shoulders, upper arms, and hips. Although GCA and PMR are associated, PMR is not known to induce visual loss. The highest incidence and prevalence occur in Northern Europe [2]. It is estimated that, by 2050, approximately 500,000 people will have visual impairment due to GCA worldwide [3].

A direct, non-invasive, in vivo assessment of the retinal metabolism [4,5,6,7,8] and structure [8,9,10,11,12] can be conducted using a retinal oximeter, and the measurement of retinal saturation [13] and retinal vessel diameters [14] has shown acceptable repeatability.

In newly diagnosed GCA patients, retinal oximetry has shown an altered venular oxygen saturation and differences in arterio-venular oxygen saturation, even in GCA patients with no ocular symptoms [15]. However, to our knowledge, retinal oximetry or retinal vascular calibers have never been validated for application in PMR patients and have not been validated for GCA patients at 6–24 months after diagnosis. 

Thus, we aimed to investigate the vascular system in terms of the retinal metabolism and retinal vascular calibers in GCA and PMR patients at 6–24 months after diagnosis. Our objectives were to examine if (a) the retinal arteriolar oxygen saturation, (b) retinal venular oxygen saturation, (c) difference in retinal arterio-venular oxygen saturation, (d) retinal arteriolar vessel diameter, and (e) retinal venular vessel diameter differed similarly between GCA and PMR patients, on one hand, and control participants, on the other hand. 

## 2. Materials and Methods

### 2.1. Study Design

This clinical cross-sectional study was conducted between March 2021 and January 2022. GCA and PMR patients with the International Classification of Diseases (10th revision) and Related Health Problems (ICD-10) codes *M315, M315A, M316, M316A (GCA),* and *M353 (PMR)* were identified in medical journal files from the Department of Rheumatology at Odense University Hospital, Odense, Denmark, or Svendborg Sygehus, Svendborg, Denmark, and invited to participate in the study via e-mail.

We divided patients into a group of GCA patients and a group of PMR patients.

GCA patients were included if a positive temporal artery biopsy, positive positron emission tomography/computer tomography scans, or positive ultrasonography were obtained, as described in the 2018 European League Against Rheumatism recommendations [16]. 

PMR patients with a score of four or above on the European League Against Rheumatism 2012 classification criteria were included. Age above 50 years, bilateral shoulder pain, and elevated C-reactive protein (CRP) were mandatory, while negative anti-citrullinated protein antibodies (Anti-CCP) and IgM rheumatoid factor (IgM-RF) scored two points and morning stiffness >45 min scored two points. Pelvic pain and no swollen or painful distal joints each scored one point [17]. Furthermore, the GCA and PMR patients had to be diagnosed between 6–24 months before inclusion. A senior specialist in rheumatology (KEB) verified the diagnosis. We excluded images with a quality below 6.0 (range 0–10) as graded by the Oxymap T1 (Oxymap, Reykjavik, ehf.). Moreover, we excluded patients with neurodegenerative disease, including Parkinson’s disease, Alzheimer’s disease, multiple sclerosis, and patients with retinal or optic nerve diseases, including glaucoma, larger retinal oedemas, arteritic anterior ischemic optic neuropathy (AAION), non-arteritic anterior ischemic optic neuropathy (NAION), macular degeneration diseases, retinal tears, retinal detachment, macular holes, and retinitis pigmentosa. Furthermore, we excluded patients with diabetic retinopathy, which can affect retinal non-invasive measurements [18]. Yet, minimal small retinal drusen and minor epiretinal fibrosis were not regarded as ophthalmologic diseases. The time of diagnosis of GCA and PMR was defined as the date when the patient received treatment, and the symptom debut was defined as the first time a patient described symptoms of disease.

Control participants were invited the Department of Ophthalmology, Odense University Hospital, Odense, Denmark, prior to cataract surgery and recruited after surgery if no rheumatologic, cancer or relevant retinal or optic nerve diseases were present at the time of inclusion. Minimal small retinal drusen and minor epiretinal fibrosis were not considered ophthalmologic diseases. 

### 2.2. Variables

Our predictors were the study groups (GCA patients and PMR patients, respectively).

The outcomes of the study comprise retinal arteriolar oxygen saturation, a measure of the mean saturation in the largest arteriole of each quadrant of the retina, given as a percentage; retinal venular oxygen saturation, a measure of the mean saturation in the largest venule of each quadrant of the retina, given as a percentage; the difference in retinal arterio-venular oxygen saturation, a measure of the difference between the mean arteriolar oxygen saturation and the mean venular oxygen saturation, given as a percentage; retinal arteriolar vessel diameter, a measure of the mean vessel diameter in the largest arteriole of each quadrant of the retina, given in micrometers; and retinal venular vessel diameter, a measure of the mean vessel diameter in the largest venule of each quadrant of the retina. All means are based on three or four quadrants.

Confounders include age and sex, which were adjusted for in the linear mixed regression model. 

We do not believe that any variable had a different effect on the GCA group compared to PMR group, respectively. Thus, we assume that no effect modifiers were present in this study.

### 2.3. Clinical Examinations

We carried out all clinical examinations at the Department of Ophthalmology at Odense University Hospital, Odense, Denmark. The CRP value closest to inclusion was defined as the latest blood sample drawn before the study visit.

We obtained medical history for diagnoses with giant cell arteritis and polymyalgia rheumatica. Moreover, based on medical journals over a period of five years prior to inclusion, participants were scored according to the Charlson comorbidity index (CCI), using ICD-10 [19]. The CCI score is based on 19 different comorbidities added to a score based on the age of the patient, hence providing one overall score. As the CCI scores of GCA and PMR patients automatically included one point for connective tissue disease, an adjusted CCI was created so as to compensate by subtracting one point from the GCA and PMR patients, respectively. 

Upon the hospital visit, the blood pressure was obtained in mmHg and measured three consecutive times from the left arm using the same monitor. Mean systolic blood pressure (SBP) and diastolic blood pressure (DBP) was calculated. Mean arterial blood pressure (MAP) was defined as *DBP + 1/3(SBP − DBP)*. Height was measured in meters. Weight was measured in kilograms with clothes. Body mass index (BMI) was calculated as height/(meters^2^). The ophthalmological examination consisted of ophthalmoscopy, with the best-corrected visual acuity tested using the Early Treatment Diabetic Retinopathy Study chart at four meters (Precision Vision, Woodstock, IL, USA). Intraocular pressure was measured using Goldmann applanation tonometry and was provided in mmHg. Pupils were dilated using tropicamide 1% and supplemented with phenylephrine 10% if required for optimal dilation. 

### 2.4. Retinal Oximetry and Image Analysis

We used Oxymap T1 (Oxymap ehf., Reykjavik, Iceland) to detect the retinal arteriolar, venular oxygen saturation, and retinal blood vessel calibers. The Oxymap T1 device comprises two digital cameras and an optical adapter attached to a TRC-50DX fundus camera (Topcon Corporation, Tokyo, Japan). Fundus images were captured using a 50° view at wavelengths of 570 nm and 600 nm of light simultaneously. The fact that light absorbance by the blood affects the brightness in the blood vessels, but not the sides of the blood vessels, is useful for finding the optical density at each wavelength. The ratio between the optical densities of each wavelength is inversely and linearly related to the retinal oxygen saturation [20]. 

The optic disc was centered in the image, and at least two pictures were captured of each eye, starting with the right eye. The image with the highest overall quality was chosen for further analysis. The image quality was automatically graded on a scale from 0 to 10 by the Oxymap T1 (Oxymap ehf.). The pictures were taken at different light intensities, starting with no light and gradually building up the light intensity for each image. A circle was manually fitted to the optic disc, and two more circles were semi-automatically enlarged to 0.5 times and 3.0 times the size of the circle surrounding the optic disc, respectively. The circle surrounding the optic disc was deleted, and blood vessel measurements were performed manually in the space between the two remaining circles. The required length of the blood vessels for inclusion was between ≥50 pixels and ≤200 pixels. The retinal arteriole and venule with the most considerable length and diameter were chosen from each quadrant, starting with the upper nasal quadrant, and followed by the lower nasal quadrant, the lower temporal quadrant, and the upper temporal quadrant. Mean values for the retinal arteriolar and venular oxygen saturation and blood vessel diameters were calculated.

All images were captured and analyzed by S.J.L. For their conversion from pixels to µm, a conversion factor of 9.3 was used [14]. A retinal oximetry fundus image is graphically depicted in Figure 1 to illustrate the methods. Retinal oximetry is a subclinical measurement, which is not suitable for direct in vivo clinical interpretation. All images were analyzed using Oxymap Analyzer Software 2.5.0 (Oxymap ehf., Reykjavik, Iceland). 

### 2.5. Ethics

The tenets of the Declaration of Helsinki were followed, and the study was approved by the Record of Data Processing Activities in the Region of Southern Denmark (20/61779) and by the Research Ethics Committee in the Region of Southern Denmark (20200189). All participants in this study gave informed consent and were allowed to withdraw their consent at any time. Participation in the study was non-invasive and without risk for the participants. However, the dilation of the pupils could burden the participants with blurred vision for 4–6 h afterwards.

### 2.6. Statistical Methods

For the presentation of the demographic data, categorical variables are presented as counts (n) and proportions (%). Continuous variables are reported as medians with interquartile ranges (IQR). For the comparison of the continuous variables, the Kruskal–Wallis test and Wilcoxon rank sum test were used, as the data were not normally distributed. Pearson’s chi-squared test was used for the categorical data. 

As every participant could contribute two eyes, the cluster robust standard error was used in a linear mixed model regression analysis, adjusted for age and sex, to determine the retinal arteriolar and venular oxygen saturation, the arteriolar–venular difference, and the retinal arteriolar and venular vessel diameter. All tests were two-sided, and *p*-values of <0.05 and 95% confidence intervals that were not null were considered statistically significant. Regarding the regression analysis, there were no missing data among the outcome-variables, including the mean retinal arteriolar and venular oxygen saturation, arterio-venular difference, and mean retinal arteriolar and venular vessel diameters (mean values were based on data of three to four retinal quadrants).

Statistical analyses were carried out in Stata 17 (StataCorp, College Station, TX, USA). 

The sample size was estimated from the findings by Türksever et al. [15], according to which the retinal vascular oxygen saturation was determined for the newly diagnosed patients and control participants. On the assumption that PMR patients have the same arteriolar–venular difference as GCA patients, at 32.2% ± 3.8%, and that the control participants have an arteriolar–venular difference of 38.3% ± 2.8%, we deemed that we would require 12 patients in each group (α = 0.05, power 0.90). 

## 3. Results

### 3.1. Patient Population

We invited 194 patients with either GCA (n = 40) or PMR (n = 154) to participate in the study (Figure 2). The reasons for non-attendance were no response (n = 98) and the fact that the patient was non-contactable by e-mail (n = 36) or did not want to participate (n = 12), and nine participants were excluded for medical reasons, including glaucoma (n = 1), Alzheimer’s disease (n = 1), dry age-related macular degeneration (n = 3), poor image quality under 6.0 (n = 2), bilateral retinal detachment (n = 1), or because they did not fulfill the 2012 European League Against Rheumatism criteria for PMR (n = 1). Two GCA patients participated with a single eye due to former retinal detachment (n = 1) and severe epiretinal fibrosis with lamellar macular holes (n = 1). Among the PMR patients, three participated with a single eye due to poor image quality (n = 2) or retinal tear (n = 1). 

Of the control participants, thirty-three responded to the invitation, of whom eleven rejected participation and ten were excluded due to highly suspected glaucoma (n = 1), large retinal oedema (n = 1), dry age-related macular degeneration (n = 2), pseudovitelliform macular degeneration (n = 2), a former PMR diagnosis (n = 1), diabetic retinopathy (n = 1), disseminated colon cancer (n = 1), and small lymphocytic lymphoma (n = 1). Of the remaining 12, four participated with a single eye due to poor image quality (n = 1), highly suspected glaucoma (n = 1), severe epiretinal fibrosis (n = 1), and postoperative cystoid macular edema after cataract surgery (n = 1). 

Thus, we included 20 participants (38 eyes) with GCA, 19 participants (35 eyes) with PMR, and 12 control participants (20 eyes). 

### 3.2. Clinical Characteristics

Table 1 demonstrates an overview of the demographic parameters of each group, including the number of individuals per group, stratified by age. Among all participants, the median age (IQR) was 70.0 years (67.0 to 77.0), but the control participants were older compared to the GCA and PMR patients. Twenty participants (39%) were male and twenty participants (39%) were cataract-operated (pseudophakic) in both eyes, including all control participants. No difference was found in the body mass index, systolic and diastolic blood pressure, and likewise in the mean arterial blood pressure between any of the groups.

While patients with GCA and PMR did not differ in any of the parameters, the controls had a higher age (*p* = 0.039), higher image quality (*p* = 0.002), higher best-corrected visual acuity (*p* = 0.017), and lower intraocular pressure (*p* = 0.008). Additionally, a difference in the Charlson comorbidity index (*p* < 0.001) was found between all groups but not in the adjusted Charlson comorbidity index (*p* = 0.54). 

### 3.3. Retinal Metabolism and Structure

An overview of the linear mixed regression model analysis adjusted for age and sex is found in Table 2. The GCA and PMR patients differed from control participants in the retinal arteriolar diameter (124.4 µm vs. 115.3 µm, *p* = 0.023 and 124.8 µm vs. 115.3 µm, *p* = 0.049, respectively).

There were no differences between any of the groups with respect to the retinal arteriolar oxygen saturation, retinal venular oxygen saturation, arterio-venular difference, or retinal venular diameter.

## 4. Discussion

This cross-sectional study aimed to investigate whether the vascular system in patients with GCA and PMR differed similarly in terms of the retinal metabolism and retinal vascular calibers compared to control participants. The results showed a significantly wider arteriolar diameter in GCA patients compared to control participants and in PMR-patients compared to control participants. This may indicate a microvascular overlap between GCA and PMR. 

The retinal vessel diameter can be adjusted by a complex autoregulatory system [21] and has been shown to decrease during breathing with 100% oxygen [9,11]. The increase in the arteriolar diameter among patients with GCA and patients with PMR compared to the control participants in our study may be caused by high metabolic demands or, perhaps, irregularities in the autoregulatory metabolic system, as observed in diabetic retinopathy [22]. 

Patients with ocular and central nervous system sarcoidosis showed a difference in the retinal arterio-venular oxygen saturation compared to sarcoidosis patients whose ocular and central nervous systems were not affected. However, no difference was found in the retinal venular vessel diameter or arteriolar vessel diameter [8]. Accordingly, newly diagnosed GCA patients were investigated using retinal oximetry by Türksever et al. [15]. In contrast to our results, they found that the retinal arterio-venular difference decreased, and the retinal venular oxygen saturation increased significantly compared to the control participants. However, the retinal vessel calibers were not measured. 

The dilation of the retinal venular diameters has previously been linked to inflammation [23]. Although we did not find altered retinal venular calibers in the GCA and PMR patients compared to the control participants, inflammation cannot be ruled out as a potential cause of the retinal arteriolar widening found in our results. 

In patients with Voigt–Koyanagi–Harada disease, immunosuppressive treatment decreased the retinal arteriolar and venular diameter at follow-up compared to the baseline [10]. The majority of GCA and PMR-patients in our study had already received corticosteroid treatment. Corticosteroid, in larger doses, is a well-established treatment for GCA and PMR, with good treatment responses [24], which may have influenced the retinal arteriolar- and venular oxygen saturation, the retinal arterio-venular difference in oxygen saturation, and the retinal venular diameter between groups in our study.

Strengths can be found in this study. Firstly, retinal examinations were conducted to eliminate the risk of significant ocular disease. Secondly, this was, to our knowledge, the first study to investigate retinal metabolism and the retinal structure in patients with GCA and PMR at 6–24 months after diagnosis. Finally, the image capturing and analysis were performed by the same investigator (SJL). Limitations can also be found within this study. Firstly, it was a cross-sectional study and, hence, causality cannot be determined. Secondly, we were only able to include a limited number of patients and, thus, the risk of type-2 errors must be considered when interpreting the results. Thirdly, some clinical characteristics differed between the control participants and GCA and PMR patients, potentially affecting the results. Finally, we cannot conclude whether the same patterns would emerge in newly diagnosed treatment-naïve patients. 

## 5. Conclusions

In conclusion, we found a wider arteriolar diameter comparing the GCA and PMR patients to the controls. These results suggest that GCA and PMR patients differ similarly in terms of their retinal structure compared to controls at 6–24 months after diagnosis. Prospective studies on newly diagnosed treatment-naïve patients could help us to determine whether changes in the retinal metabolism and structure can be used predictively or whether they simply reflect the initiated treatment.

## Figures and Tables

**Figure 1 metabolites-12-00872-f001:**
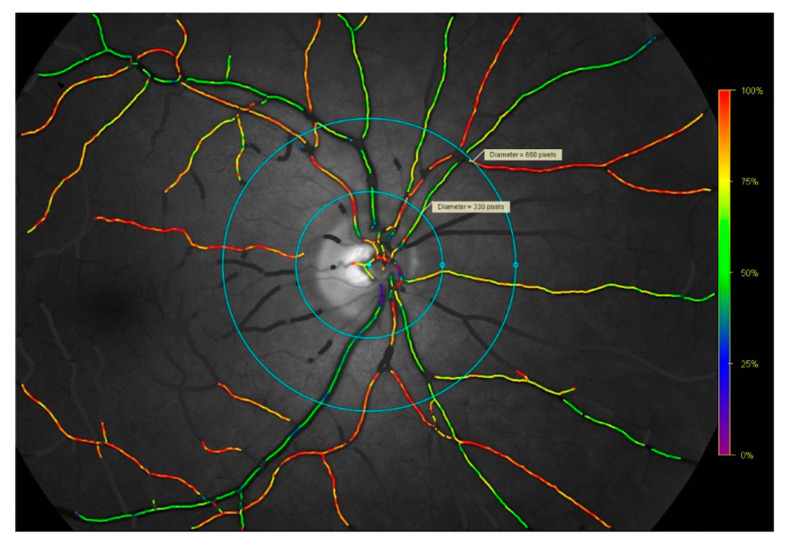
Retinal oximetry fundus image captured with the TRC-50DX fundus camera and Oxymap T1 oximeter device. Example of a retinal oximetry fundus image depicting the right eye of a patient with giant cell arteritis, included here to illustrate the methods. Images were analyzed using the Oxymap 2.5.0. Oxymap Analyzer software (Oxymap ehf., Reykjavik, Iceland) in the space between the inner and outer circles for the assessment of the arterioles and venules of each quadrant. Purple, blue, and green vessels reflect a lower saturation in the range of approximately 0–65%. Yellow, orange, and red vessels reflect a higher saturation in the range of approximately 66–100%. Retinal oxygen saturation and retinal vessel calibers were manually measured in the space between the inner and outer light-blue circles.

**Figure 2 metabolites-12-00872-f002:**
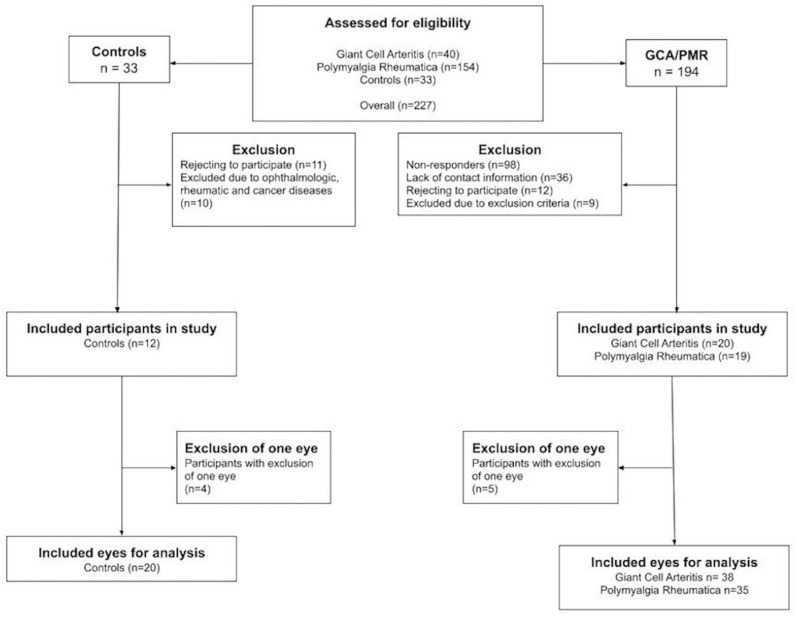
Flowchart depicting the inclusion process for patients with giant cell arteritis, polymyalgia rheumatica, and control participants.

**Table 1 metabolites-12-00872-t001:** Clinical characteristics of the controls, giant cell arteritis patients, and polymyalgia rheumatica patients.

		Control Participants	GCA Patients	PMR Patients	*p*-Value	*p*-Value ^††^
Individuals overall, (n)		12	20	19		
Age +80 years, n (%)		4 (33%)	1 (5%)	1 (5%)		
Age 71–80 years, n (%)		5 (42%)	8 (40%)	5 (26%)		
Age 61–70 years, n (%)		3 (25%)	11 (55%)	11 (58%)		
Age 50–60 years, n (%)		0 (0%)	0 (0%)	2 (11%)		
Age at visit (years), median (IQR)		75.5 (71.5 to 81.0)	69.0 (66.5 to 76.5)	69.0 (67.0 to 72.0)	0.039 ^†^*	0.58 ^§^
Sex, n (%)	Male	3 (25%)	7 (35%)	10 (53%)	0.27 ^‡^	0.27 ^‡^
Body mass index (weight/(height^2^)), median (IQR)		25.6 (23.0 to 29.5)	25.6 (24.2 to 29.4)	28.4 (23.9 to 31.5)	0.65 ^†^	0.50 ^§^
Systolic blood pressure (mmHg), median (IQR)		149.0 (130.0 to 172.5)	150.5 (137.5 to 159.5)	144.0 (125.0 to 156.0)	0.53 ^†^	0.29 ^§^
Diastolic blood pressure (mmHg), median (IQR)		85.5 (82.5 to 97.0)	89.5 (82.0 to 94.5)	90.0 (76.0 to 94.0)	0.92 ^†^	0.70 ^§^
Mean arterial blood pressure (mmHg), median (IQR)		111.2 (98.5 to 117.7)	109.5 (104.0 to 115.8)	110.0 (94.0 to 113.0)	0.59 ^†^	0.37 ^§^
Calculated Charlson score, median (IQR)		3.0 (3.0 to 3.5)	4.0 (4.0 to 5.0)	4.0 (3.0 to 5.0)	<0.001 ^†^***	0.70 ^§^
Adjusted Charlson score for GCA and PMR, median (IQR) ^‡‡^		3.0 (3.0 to 3.5)	3.0 (3.0 to 4.0)	3.0 (2.0 to 4.5)	0.54 ^†^	0.70 ^§^
Prednisolone dose at visit (mg), median (IQR)			7.5 (3.5 to 10.0) ^‖^	5.0 (2.5 to 8.8) ^¶^		0.37 ^§^
CRP value closest to inclusion (mg/L), median (IQR)			3.0 (1.4 to 5.0)	3.4 (1.0 to 7.5)		0.46 ^§^
Time from symptom debut to diagnosis (months), median (IQR)			1.7 (0.9 to 3.7)	2.6 (1.1 to 4.3)		0.19 ^§^
Time from diagnosis to inclusion (months), median (IQR)			11.3 (6.6 to 19.2)	9.7 (6.2 to 18.5)		0.91 ^§^
Age at time of diagnosis (years), median (IQR)			68.5 (65.0 to 76.0)	68.0 (66.0 to 71.0)		0.45 ^§^
Pseudophakic in both eyes, n (%)	Yes	12 (100%)	5 (25%)	3 (16%)	<0.001 ^‡^***	0.45 ^‡^
Eyes, (n)		20	38	35		
Best-corrected visual acuity (ETDRS), median (IQR)		84.5 (84.0 to 90.5)	82.0 (77.0 to 85.0)	84.0 (81.0 to 87.0)	0.017 ^†^*	0.15 ^§^
IOP (mmHg), median (IQR)		13.0 (11.6 to 14.0)	16.0 (13.5 to 18.7) ^§§^	15.6 (10.5 to 17.8)	0.002 ^†^**	0.18 ^§^
Image quality (0 to 10), median (IQR)		8.1 (7.7 to 8.5)	7.4 (6.9 to 7.9)	7.6 (6.8 to 8.1)	0.002 ^†^**	0.28 ^§^

Variables are reported as medians and interquartile ranges (IQR), counts (n), and proportions (%). ^†^ Kruskal–Wallis test. ^‡^ Pearson’s chi-square test. ^‡‡^ Charlson comorbidity index score adjusted for GCA and PMR by the subtraction of one point to compensate for connective tissue disease. ^§^ Wilcoxon rank sum test. ^‖^ Two did not receive treatment. ^¶^ Six did not receive treatment. ^††^ Giant cell arteritis compared to polymyalgia rheumatica. ^§§^ One participant did not have their intraocular pressure measured. IOP: intraocular pressure. ETDRS: Early Treatment Diabetic Retinopathy Study chart. GCA: giant cell arteritis. PMR: polymyalgia rheumatica. * *p*-value < 0.05. ** *p*-value < 0.01. *** *p*-value < 0.001.

**Table 2 metabolites-12-00872-t002:** Differences between patients with giant cell arteritis and polymyalgia rheumatica against the controls.

	Control Participants	GCA Patients	*p*-Value ^†^	PMR Patients	*p*-Value ^‡^
Retinal arteriolar oxygen saturation, % (SE)	91.3 (0.92)	92.9 (0.86)		93.4 (0.87)	
Difference compared to controls (%), β (95%CI)	Reference	1.53 (−0.79 to 3.85)	0.20	2.02 (−0.57 to 4.61)	0.13
Retinal venular oxygen saturation, % (SE)	55.0 (2.20)	53.7 (2.30)		53.0 (2.04)	
Difference compared to controls (%), β (95%CI)	Reference	−1.29 (−7.52 to 4.93)	0.68	−2.04 (−8.32 to 4.24)	0.52
Difference in retinal arterio-venular saturation, % (SE)	36.3 (1.79)	39.1 (1.83)		40.4 (1.92)	
Difference compared to controls (%), β (95%CI)	Reference	2.80 (−2.38 to 7.98)	0.29	4.05 (−1.44 to 9.54)	0.15
Retinal arteriolar vessel diameter, µm (SE)	115.3 (3.03)	124.4 (2.40)		124.8 (3.68)	
Difference compared to controls (µm), β (95%CI)	Reference	9.14 (1.24 to 17.05)	0.023 *	9.54 (0.03 to 19.05)	0.049 *
Retinal venular vessel diameter, µm (SE)	155.1 (4.02)	162.0 (2.51)		160.1 (3.11)	
Difference compared to controls (µm), β (95%CI)	Reference	6.83 (−1.80 to 15.43)	0.12	4.97 (−5.00 to 14.94)	0.33

Retinal saturation is given as percentages. Retinal vessel diameters are given as µm. Retinal oxygen saturation and vessel diameters are predicted values. ^†^ Comparison of GCA against controls. ^‡^ Comparison of PMR against controls. Note: β: coefficient of the linear mixed model regression; 95% CI: 95% confidence interval; GCA: giant cell arteritis; PMR: polymyalgia rheumatica. * *p*-value < 0.05.

## Data Availability

The data that support the findings of this study are available from the corresponding author upon reasonable request, with the acceptance of Research Ethics Committee in the Region of Southern Denmark.

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
