# Peer review of "Non-Invasive Metabolic and Structural Retinal Markers in Patients with Giant Cell Arteritis and Polymyalgia Rheumatica: A Cross-Sectional Study"

_metabolites, 2022, doi:10.3390/metabo12090872_

Round 1
Reviewer 1 Report
The study entitled "Non-invasive metabolic and structural retinal markers in pa- 2 tients with giant cell arteritis and polymyalgia rheumatica: A 3 cross-sectional study." is a well written and interesting study. Authors found no difference of SO2 between groups. The issue is original and well presented.
Author Response
Dear Reviewer,
We thank you very much for taking the time to review our manuscript.
Reviewer 2 Report
Giant cell arteritis is an inflammation of the lining of your arteries. The current study aimed to investigate the vascular system in term of retinal metabolism.
1. The number of patients is less, I would suggest including more patient.
2. The author must include fundus image of controls too, so that the reader can distinguish the retinal vasculature between patient and controls.
3. I would suggest to include the level of inner retinal oxygen delivery in the study.
This study is a case control study and I would suggest this study to transfer in a journal specific to vision opthalmology in patients.
Author Response
Dear Reviewer,
Thank you very much for your comments and for taking time to review our manuscript.
We have tried to answer your comments in the best possible way.
Question 1: The number of patients is less, I would suggest including more patients.
Answer 1: This is a very good and relevant point. The sample size was estimated based on a previous study by Türksever et al. and, regretfully, it was not possible to include additional patients, as we had invited all 194 patients with GCA and PMR registered in our database with the time of diagnosis at 6-24 months prior to inclusion. We have mentioned this among the limitations of the study.
Question 2: The author must include fundus image of controls too, so that the reader can distinguish the retinal vasculature between patients and controls.
Answer 2: We fully understand this relevant comment. However, since the picture presented in the article is only meant for illustration of our methods, we did not include a picture of a control participant. In the revision, we have tried to emphasize that retinal oximetry is a subclinical measurement, which is not available for direct in-vivo clinical interpretation (under the heading “Retinal oximetry and Image Analysis in “Materials and Methods” (Line 161-162) and in the figure-text for Figure 1 (Line 167).
Question 3: I would suggest to include the level of inner retinal oxygen delivery in the study.
Answer 3: This would have been a very interesting and relevant approach that could supplement our results in a positive way. Unfortunately, we did not measure flow rate in any patients using Doppler Technique, as this was not the aim of the study.
Question 4: This study is a case-control study and I would suggest this study to transfer in a journal specific to vision ophthalmology in patients.
Answer 4: We understand this point. However, we do find this journal relevant for this multidisciplinary dataset that we present here.
Reviewer 3 Report
This is a very interesting cross-sectional study comparing retinal oxygen saturation and retinal vessel calibers in GCA patients, PMR patients, and the control group. I consider this manuscript to be competently written, broadly following the SROBE guidelines. The introduction is brief but very to the point, presenting the scientific background and justification of the investigation. The results presented also support the conclusions. I consider that the statistical analysis is correctly performed
However, there two minor issues in this manuscript that the authors could address:
-In the methods section please present the variables, clearly define all outcomes, exposures, predictors, potential confounders and effect modifiers.
-In the results section clinical and demographic characteristics should be presented in more detail
Author Response
Dear Reviewer,
Thank you very much for your comments and for taking time to review our manuscript.
We have tried to answer your comments in the best possible way.
Question 1: In the methods section please present the variables, clearly define all outcomes, exposures, predictors, potential confounders, and effect modifiers.
Answer 1: This is a highly relevant point, and we have added the following information to the article. We also have a new section “Variable” (Starting at Line 80 - 110) in the “Materials and Methods”
Predictors:
Our predictors in this study are GCA-group and PMR-group, respectively. (Line 81)
Outcomes:
Outcomes of the study comprise retinal arteriolar oxygen saturation, retinal venular oxygen saturation, retinal arterio-venular oxygen saturation difference, retinal arteriolar vessel diameter, and retinal venular vessel diameter. We have added the information in lines 46-49 and the section “Variable” (Line 100 - 107)
Confounders:
In the “materials and methods-section” under the heading “Statistical methods” we have now stated in the text that age and sex are regarded as confounders and are thus, adjusted for in the linear mixed regression model. (Line 108)
Effect Modifiers:
We do not believe that any variables differ in effect across the GCA-group and PMR-group and thus, we assume that no effect modifiers are present in this study. (Line 109 - 110)
In the “Materials and methods-section” under the heading “Study-design” we defined patients with neurodegenerative disease as Parkinsons disease, Alzheimers disease and multiple sclerosis (Line 67 - 68).
Additionally, in the same section (Materials and methods) and under the same heading (Study-design) we defined retinal- and optic nerve disease as glaucoma, larger retinal oedema, arteritic anterior ischemic optic neuropathy (AAION), non-arteritic anterior ischemic optic neuropathy (NAION), pseudovitelliform- and age-related macular degeneration (AMD), retinal tear, retinal detachment, macular hole, retinitis pigmentosa and furthermore diabetic retinopathy (Line 68 - 72).
In the “materials and methods-section” under the heading “Study population”, Time of diagnosis was defined as the date the patient received treatment and symptom-debut was defined as the first time a patient described symptoms from disease (Line 73 - 74).
In the “materials and methods-section” under the heading “Clinical Examinations” we defined the Charlson Comorbidity Index (CCI). Based on medical records from a period of five years prior to inclusion participants were scored on CCI using ICD-10. The CCI is based on 19 different comorbidities plus the age of the patient providing one overall score. As the CCI-score of GCA and PMR-groups automatically scored one point for the connective tissue disease an adjusted CCI was made for compensation by subtracting one point from GCA- and PMR-groups, respectively (Line 115 - 119).
In the “materials and methods-section” under the heading “Clinical Examinations” we defined blood pressure, height, weight, and BMI.
Blood pressure was obtained in mmHg and measured three consecutive times on the left arm using the same monitor for calculation of a mean systolic blood pressure (SBP) and diastolic blood pressure (DBP). Mean arterial blood pressure (MAP) was defined as DBP +1/3(SBP - DBP).
Height was measured in meters. Weight was measured in kilograms with clothes. Body mass index (BMI) was calculated as height/(meters2) (Line 120 - 123).
Furthermore, CRP-value closet to inclusion was defined as the latest blood sample drawn before the study examination and likewise was described under this heading (Line 114).
In the “materials and methods-section” under the heading “Clinical Examinations” we defined Best-corrected visual acuity as tested using the Early Treatment Diabetic Retinopathy Study chart at four meters (Precision Vision, Woodstock, IL, USA).Pupils were dilated using tropicamide 1% and supplemented with phenylephrine 10%, if needed for optimal dilation.
Image quality was automatically graded on a scale from 0 to 10 by the Oxymap T1 (Oxymap ehf.).
Intraocular pressure was measured using Goldmann applanation tonometry and was provided in mmHg (Line 123 - 127).
Question 2: In the results section clinical and demographic characteristics should be presented in more detail.
Answer 2: This is also a very relevant point. Thus, we have added the following to Table 1:
Charlson Comorbidity Index Score, the Charlson Comorbidity Index minus one point for connective tissue disease (=1 point subtracted from GCA-group and PMR-group, respectively), Body Mass Index (= weight in kilograms/(height^2), Age at time of diagnosis in years of age and Number of Individuals stratified by age-groups (50-60, 61-70, 71-80, +80 years). Furthermore, we have explained Table 1 a little more in detail in the text under the heading “Clinical Characteristics” in “Materials and Methods”.
Round 2
Reviewer 2 Report
The author has modified the paper and can be accepted for publishing.